# A Diagnostic Model to Predict SARS-CoV-2 Positivity in Emergency Department Using Routine Admission Hematological Parameters

**DOI:** 10.3390/diagnostics11091566

**Published:** 2021-08-28

**Authors:** Rossana Soloperto, Giovanna Guiotto, Giuseppe Tozzi, Maurizio Fumi, Angelo Tozzi

**Affiliations:** 1Department of Advanced Biomedical Science, Federico II University Hospital, Via S. Pansini 5, 80131 Naples, Italy; 2Department of Emergency Medicine, AORN San Pio, Via dell’Angelo 1, 82100 Benevento, Italy; tozziangelo@libero.it; 3Department of Statistical and Actural Science, Sannio University, Via Traiano 3, 82100 Benevento, Italy; giuseppe98napoli@hotmail.it; 4Clinical Pathology Laboratory, AORN San Pio, Via dell’Angelo 1, 82100 Benevento, Italy; maurizio.fumi@tiscali.it

**Keywords:** COVID-19, laboratory diagnostics, predictive model, routine blood tests, SARS-CoV-2

## Abstract

Early detection of SARS-CoV-2 in the emergency department (ED) is a crucial necessity, especially in settings of overcrowding: establishing a pre-diagnostic test probability of infection would help to triage patients and reduce diagnostic errors, and it could be useful in resource-limited countries. Here, we established and validated a clinical predictor of infection based on routine admission hematological parameters. The diagnostic model was developed by comparing 85 consecutive patients with symptomatic COVID-19 confirmed by RT-PCR with 85 symptomatic, SARS-CoV-2-negative controls. Abnormal hematological parameters significantly (*p* < 0.05) associated with SARS-CoV-2 infection were used to derive a “cumulative score” between 0 and 16. The model was validated in an independent cohort of 170 SARS-CoV-2-positive patients. Several routine hematology parameters were significantly (*p* < 0.05) associated with SARS-CoV-2 infection. A “cumulative score” score ≥7 discriminated COVID-19-postive patients from controls with a sensitivity of 94% and specificity of 100% (*p* < 0.001). The high sensitivity of the predictive model was confirmed in the prospective validation set, and the cumulative score (i) predicted SARS-CoV-2 positivity even when the first oro-nasopharyngeal swab RT-PCR result was reported as a false negative in both cohorts and (ii) resulted to be independent from disease severity. The cumulative score based on routine blood parameters can be used to predict an early and accurate diagnosis of SARS-CoV-2 infection in symptomatic patients, thereby facilitating triage and optimizing early management and isolation from the COVID-19 free population, particularly useful in overcrowding situations and in resource-poor settings.

## 1. Introduction

The COVID-19 pandemic caused by the SARS-CoV-2 virus continues as a serious pandemic. Early and rapid diagnosis remains a priority to identify and isolate infected patients, especially in overcrowded emergency department (ED). While real-time (RT-PCR) analysis of specimens collected through oro-nasopharyngeal swabs is the gold standard COVID-19 diagnostic assay, the sensitivity and specificity of this approach is not 100% [1]. Furthermore, in developing countries, the availability of this test is often limited. To minimize false positive and negative results, RT-PCR results must be interpreted according to a pre-test probability of infection established clinically [2]. However, there are no specific routine indicators of COVID-19 to help with this decision-making.

Several studies and recent metanalyses [3,4] have reported associations between ongoing SARS-CoV-2 infection and abnormal laboratory test results. Hypoalbuminemia [5,6,7,8], lymphocytopenia [8,9,10,11], neutrophilia [8], increased lactic dehydrogenase (LDH) [5,8], C-reactive protein (CRP) [8,9], procalcitonin [5,9,10], D-dimers [7,11,12,13], and neutrophil-lymphocyte ratio (NLR) [14,15] levels on admission or in serial measurements over time have all been reported as predictors of the severity of infection and have been associated with poor outcomes, intensive care unit admission, severity of pneumonia assessed by computed tomography (CT) [16], or the extent of inflammation or tissue damage [17]. However, most of these studies have focused on prognosis rather than the diagnostic value of blood test abnormalities. 

In order to develop a specific diagnostic algorithm for SARS-CoV-2 positivity based on routine admission hematological parameters, here, we compared the hematological results of COVID-19 patients and controls obtained within the first 24 h of admission to the ED. The sum of the number of all altered blood test results, which we call the “cumulative score”, provided a specific and sensitive pre-test probability of SARS-CoV-2 infection. The cumulative score may be useful for optimizing triage and management, reducing missed diagnoses, and excluding infection in symptomatic patients admitted with other COVID-19 mimics. Furthermore, we validated the predictive score in an independent prospective cohort of COVID-19 patients, confirming its high sensitivity, and we analyzed its correlation with COVID-19 disease severity.

## 2. Materials and Methods

### 2.1. Patients, Inclusion and Exclusion Criteria, and Data Extraction

This was a single-center, retrospective case-control analysis of consecutive 85 COVID-19-positive patients and 85 matched controls admitted to the ED between 1 March 2020 and 24 May 2020. Inclusion criteria for COVID-19-positive patients were: (i) clinical symptoms or signs of SARS-CoV-2 infection on admission to the ED, i.e., fever within the 48 h prior to admission, and/or clinical dyspnea or respiratory failure (defined as an oxygen saturation with FiO_2_ 21% less than 90% on pulse oximetry or a pO_2_ < 60 mmHg on arterial blood gas analysis for non COPD patients), and/or gastrointestinal disorders, or pauci-symptomatic patients (flu-like or colds-like symptoms, or isolated anosmia or ageusia) showing chest CT findings (retrospectively analyzed) positive for interstitial bilateral pneumonia; and (ii) SARS-CoV-2 positivity established via RT-PCR on oro-nasopharyngeal swabs within 72 h of hospitalization. Control patients were 85 consecutive patients admitted over the same period but who subsequently had negative RT-PCR results. The inclusion criteria for control patients were: (i) COVID-19-like symptoms; and (ii) established SARS-CoV-2 negativity both by oro-nasopharyngeal swab RT-PCR and SARS-CoV-2 serum immunoglobulin testing. The exclusion criteria for both COVID-19 patients and controls were: (i) the presence on admission of severe comorbidities having a great impact on routine admission blood tests, i.e., hematological malignancies like lymphoma or leukemia, terminal renal or liver disease, chronic inflammatory disease, such as end-stage cancer or acute phase of inflammatory bowel’s diseases, or (ii) uncertain diagnosis, due to discrepancy between COVID-19 oro-nasopharyngeal swab result and serum immunoglobulin test, in order to minimize confounding factors or potential selection bias. Possible bias remains, as: a potential impact of patients’ comorbidities on blood tests and potential age-related influences, being pediatric population (minor than 18 years old) admitted in pediatric Emergency Department, and are, therefore, not present in our data. Potential confounding errors remain: intrinsic laboratory systematic errors in sample analysis; slightly hemolyzed blood samples among those analyzed. 

Data were extracted from the DNWEB hospital laboratory test results platform and from the ADT electronic medical record platform to compile all clinical and laboratory data.

The retrospective analysis resulted in a cumulative score (CS), defined as the sum of the number of altered blood tests per patient, which was used to discriminate between the two groups. The CS was prospectively validated in an independent cohort of 170 consecutive COVID-19 patients admitted to the same ED during the “second wave” of the pandemic between 21 October 2020 and 9 March 2021 using the same inclusion and exclusion criteria.

### 2.2. Statistical Analysis

All analyses were performed using IBM SPSS Statistics v20.0 (IBM Statistics, Chicago, IL, USA). All variables were described with descriptive summary statistics. For the retrospective case-control analysis, normality of distributions was checked using the Shapiro-Wilk test. Independent t-tests and Welch’s *t*-tests were used to compare the means between groups for normally distributed and equal variance or unequal variance parameters established by Levene’s test, respectively. For non-normal distributions, the non-parametric Levene’s test through rank transformation with one-way ANOVA was applied to compare means. When the variance was equal, the Mann–Whitney test was performed, otherwise the independent-sample Kolmogorov–Smirnov test for two-samples was used. 

While, for most results, standard laboratory cut-offs were applicable to stratify patients, for parameters without standardized cut-offs (neutrophil-to-lymphocyte ratio (NLR), white blood cell count to lymphocyte ratio (WLR), D-dimer/eosinophil percentage plus lymphocyte percentage ratio (DELR), and cumulative score (CS)), or where there were significant differences between groups with a lower cut-off of zero (i.e., for absolute eosinophil counts), receiver operating characteristic (ROC) curve analysis was performed to obtain a threshold (J value) to identify a potential limit able to discriminate the two groups. Results whose differences fell into normal laboratory range were excluded from further analysis even when significant, given that these results were unlikely to be useful in clinical practice. Results that were abnormal in <50% of COVID-19 patients were excluded, as well to guarantee a minimum sensitivity cut-off, albeit arbitrarily. Linear bivariate analysis using Kendall’s rank correlation was performed for all blood tests treated as dichotomized variables according to normal laboratory cut-off or to the thresholds established through ROC curves, for correlation analysis. The χ-squared test with 95% confidence intervals (CI) was used to compare frequencies between groups.

Based on the hypothesis that the probability of having a COVID-19 infection was likely to increase with an increasing number of altered blood tests, we assigned a point to each blood result defined as abnormal, thus creating the CS, which ranged from zero (less likely to have COVID-19) to sixteen (more likely to have COVID-19).

In a post-hoc analysis the whole COVID-19 positive population (85 patients form the case-control study, and 170 patients from the prospective cohort) was staged on the basis of specific criteria, integrating respiratory function, symptoms and chest CT radiological patterns, to assess the severity of COVID-19 disease at admission in ED. A non-parametric bivariate correlation was performed to establish if a significant relation was present between (i) disease severity and CS, (ii) disease severity and days from symptoms onset, (iii) CS and days from onset, and (iv) disease severity and the values of all sixteen blood tests, treated as continuous variables.

For ROC curve analysis and CS, missing values were excluded listwise, while, for descriptive statistics, frequency analysis, and linear bivariate correlations, missing values were excluded pairwise to maximize the use of data. *p*-values < 0.05 were considered statistically significant. Single-tailed *p*-values are reported only when two-tailed tests were not possible.

## 3. Results

### 3.1. Patient Demographics

For the retrospective case-control study, 138 patients presented to the ED between 1 March 2020 and 24 May 2020 with symptoms suspicious of COVID-19 or with mild flu-like symptoms associated with abnormal chest CT findings of bilateral interstitial pneumonia were eligible; 53 (38.4%) were excluded: 35 (25.36%) because of uncertain diagnosis due to: doubtful swab result without possibility to perform a bronchoalveolar lavage in ED, impossibility to obtain a certain diagnosis for suspected patients who died at admission before performing swab; and 18 (13.04%) because of abnormal laboratory values correlated with pre-exiting diseases having great impact on hematological tests. Eighty-five patients with confirmed COVID-19 infection were finally enrollable for the analysis, 14% of whom required a second nasopharyngeal or bronchoalveolar swab for definitive diagnosis. On 102 patients presented to the ED between 1 March 2020 and 24 May 2020 with the same signs and symptoms, 17 (17.3%) resulted not enrollable: 8 (8.16%) because of uncertain diagnosis due to a discrepancy between swab result and serum immunoglobulin tests performed during hospitalization, and 9 (9.18%) because of impacting pre-existing conditions, leaving 85 patients enrollable for the analysis. Chest X-rays or lung ultrasounds were used as screening for all patients admitted with suspicious symptoms. For all confirmed COVID-19 patients, chest CT scan was performed before specialistic department admission in order to decide, on the basis of radiological pattern, the most appropriate destination (low/high intensive care units), and, once admitted, the type of ventilation, according to radiological phenotypes described by Gattinoni at al. [18]. In the control group, chest CT scan was performed only when X-rays or lung ultrasound did not provide properly information for diagnosis and management of respiratory failure: radiological COVID-like patterns were observed only in those cases of alveolitis or atypical pneumonia. Chest CT scan positivity for bilateral interstitial pneumonia, reported by radiologist, included the following findings: peripheral, bilateral ground-glass opacities (GGOs), consolidations, or combination between GGOs with interlobular/intralobular septal thickening creating a “crazy paving” pattern and subpleural linear opacities. The mean of age of the entire cohort was 63 (±19), and 61% were male, and 39% were female. There were no statistically significant differences between the two groups in terms of age and sex (Pearson χ^2^ = 2.477; df = 1; two-tailed *p*-value = 0.116). All vital parameters and signs and symptoms on admission of COVID-19 patients and matched controls are summarized in Table 1. The etiology of admission in ED of control group is summarized in Appendix A.

For the prospective validation set, 186 consecutive patients presented to the ED between 21 October 2020 and 9 March 2021 with symptoms suspicious of COVID-19 or mild symptoms associated with highly suggestive radiological findings; 18 (9.7%) were excluded: 9 (4.8%) due to uncertain diagnosis and 9 (4.8%) for impacting pre-existing comorbidities, leaving 170 patients enrollable for the analysis, 12% of whom required a second nasopharyngeal or bronchoalveolar swab for diagnosis due to an initial false negative result. The mean of age of the entire cohort was 69 years (±14), and 73.5% were male and 26.5% female. All physiological variables and demographic characteristics of the prospective COVID-19 cohort are summarized in Table 2. A patients-flowchart is given in Appendix A.

### 3.2. Univariate Differences in Hematological Parameters between SARS-CoV-2-Positive and Negative Patients (Retrospective Case-Control Study)

Among 58 hematological parameters examined, 43 (74.1%) resulted significantly different between COVID-19 population and controls, as reported in univariate analysis showed in Appendix A. We excluded from further analysis those hematological parameters whose values fell into normality range, because it is useless in clinical practice (apart from eosinophils, whose differences between the two groups where remarkable), as well as those parameters found altered in less than 50% of COVID-19 population, due to poor sensibility. Net of the above considerations, sixteen blood parameters significantly different between COVID-19 patients and controls resulted eligible for further analysis: respectively, albumin (3.09 ± 0.58 vs. 3.71 ± 0.45 g/dL; *p* < 0.001); LDH (329.8 ± 136.4 vs. 209.5 ± 72.1 U/L; *p* < 0.001); CRP (6.52 ± 5.97 vs. 0.669 ± 0.87 mg/dL; *p* < 0.001); blood glucose levels (119.5 ± 34.3 vs. 107.9 ± 30.3 mg/dL; *p* = 0.006); prothrombin time percentage (75.9 ± 14.9 vs. 90 ± 11.8%; *p* < 0.001); fibrinogen (569.5 ± 197.4 vs. 329.4 ± 119.6 mg%; *p* < 0.001); d-dimers (1924 ± 3477 vs. 125.1 ± 91.6 ng/mL; *p* < 0.001); percentage neutrophils (74.6 ± 14.1 vs. 61.8 ± 9.7; *p* < 0.001); lymphocyte count (0.87 ± 0.38 vs. 2.1 ± 0.85 × 10^3^/μL; *p* < 0.001); percentage lymphocytes (14.9 ± 9.5 vs. 27.97 ± 8.3; *p* < 0.001); percentage eosinophils (0.411 ± 0.436 vs. 2.363 ± 1.725; *p* < 0.001); eosinophil counts (0.046 ± 0.093 vs. 0.178 ± 0.139 × 10^3^/μL; *p* < 0.001); blood urea nitrogen (18.45 ± 10.62 vs. 28.70 ± 24.92 mg/dL); NLR (9.44 ± 10 vs. 2.69 ± 2; *p* < 0.001); WLR (11.87 ± 11.67 vs. 4.08 ± 2.12; *p* < 0.001); and DELR (287 ± 646 vs. 4.21 ± 3.52 ng/mL/%; *p* < 0.001). Men had significantly greater lymphocytopenia than women (13.3 ± 8.5 vs. 18.6 ± 10.8; *p* = 0.038). Bar charts comparing means between the two groups are shown in Figure 1 below.

### 3.3. Establishing Cut-Offs for Non-Standard Parameters

Since the laboratory cut-off for absolute and percentage eosinophils was zero, it was not possible to differentiate patients on the basis of standard laboratory ranges, so ROC curve analysis was performed to identify a clinically useful cut-off. For percentage eosinophils, the optimal threshold was 0.75% (AUC of 0.947, standard error (SE) 0.016; asymptotic *p* < 0.001; 95%CI: 0.916–0.979; Youden’s J Index = 0.76, sensitivity 86%, specificity 90%), with lower values significantly associated with a higher probability of infection and values ≤0.25% having 100% specificity for SARS-CoV-2 infection. For absolute eosinophil counts, the optimal threshold was 0.055 × 10^3^/μL (AUC of 0.926; SE 0.022; asymptotic *p* < 0.001; 95%CI: 0.884–0.969; Youden’s J Index = 0.74; sensitivity 83.5%, specificity 90%), with lower values more suggestive of infection and values ≤0.02 × 10^3^/μL having 100% specificity.

Similarly, in the absence of validated normal laboratory ranges, ROC curve analysis was performed for NLR: the optimal threshold was 3.76 (AUC of 0.842; SE 0.034; asymptotic *p* < 0.001; 95%CI: 0.777–0.908; Youden’s J Index = 0.63; sensitivity 74%, specificity 89.3%), with higher values associated with a higher probability of infection and values ≥18 having a specificity of 100%. For WLR, the optimal cut-off was 5 (AUC of 0.928; SE 0.030; asymptotic *p* < 0.001; 95%CI: 0.870–0.986; Youden’s J Index = 0.78, sensitivity 84%, specificity 94%), with values ≥6.2 having a specificity of 100%. For DELR, the optimal threshold was 14 ng/mL/% (AUC of 0.956; SE 0.020; asymptotic *p* < 0.001; 95%CI: 0.916–0.996; Youden’s J Index = 0.84, sensitivity 84%, specificity 100%). All ROC curve performed are showed in Appendix A.

A frequency analysis of case and control groups for altered blood tests was performed with χ-squared test, whose results are summarized in Table 3. Linear bivariate analysis using Kendall’s rank correlation was performed for all blood tests to establish the strength of correlations. All laboratory tests, treated as dichotomized variables according to standard laboratory ranges and to the thresholds established by ROC curve analysis, were strongly correlated with SARS-CoV-2 infection (*p* < 0.01), as reported in Appendix A.

### 3.4. Development of a Predictive Score for SARS-CoV-2 Positivity

We hypothesized that the probability of having a COVID-19 infection was likely to be greater with an increasing number of abnormal blood tests, so a “cumulative score” (CS) derived from the sum of the number of all altered blood examinations was calculated for each patient; zero meant no abnormal tests, and sixteen meant that all blood examinations were abnormal, as described above.

As expected, the CS was significantly different between COVID-19 patients and controls (12.09 ± 3.08 vs. 2.20 ± 1.56; two-tailed *p* < 0.001; 95%CI: 8.68–11.07). ROC analysis was performed to identify a clinically useful cut-off to discriminate between the two groups. The ROC curve was performed on 37.6% of cases (69/179, with missing data excluded listwise), giving an AUC of 0.966 (SE 0.005; asymptotic *p* < 0.001; 95%CI: 0.986–1.000) and a threshold of 6.5 (Youden’s index 0.94, sensitivity 94%, specificity 100%). ROC curve analysis for CS is represented in Appendix A. Higher values were significantly associated with a higher probability of infection, and values lower than 6.5 excluded COVID-19 infection in all symptomatic patients (Pearson χ^2^ = 49.8; df = 1; asymptotic two-tailed *p* < 0.001; Phi coefficient 0.882). Distribution of CS values between controls and COVID-19 population is represented in Figure 2.

To note, the CS threshold predicted SARS-COV-2 positivity in 100% of positive patients (12/85) with a first (false negative) oro-nasopharyngeal RT-PCR result. 

### 3.5. Validation Set

Detailed descriptive statistics of 170 patients admitted to the ED for the same symptoms are shown in Table 4 below. Hypoalbuminemia was present in 82% (137/170) of patients, hyperglycemia in 75.7% (128/169); increased BUN in 73.4% (124/169); increased LDH in 90.9% (150/165); increased CRP in 98.2% (164/167); low PT% in 52.4% (87/166); increased fibrinogen in 94.2% (147/156); increased D-dimer in 66.9% (101/151); increased neutrophils percentage in 90.5% (153/169); low lymphocytes percentage in 98.8% (167/169); low eosinophils percentage (count inferior to 0.75%, according to ROC threshold found) in 91.1% (154/169); low lymphocytes count in 77.5% (131/169); low eosinophils absolute count (count inferior to 0.055 × 10^3^/μL) in 88.2% (149/169); high NLR (higher than 3.76) in 98.2% (166/169); high WLR (higher than 5) in 98.8% (167/169); high DELR (values higher than 14 ng/mL/%) in 94.5% (138/146) patients. 

The CS for SARS-CoV-2 infection was calculated in 75.3% (128/170) of patients (missing data excluded listwise), with a mean of 13.81 ± 1.58 and a median of 14. Of 128 patients, 0.78% (1/128 patients) had a CS of 8; 1.56% (2/128) had a CS of 9; 1.56% (2/128) had a CS of 10; 3.91% (5/128) had a CS of 11; 6.25% (8/128) had a CS of 12; 23.44% (30/128) had a CS of 13; 28.13% (36/128) had a CS of 14; 21.09% (27/128) had a CS of 15; and 13.28% (17/128) had a CS of 16. The distribution of CS in the prospective cohort is represented in Figure 3.

None of patients had a CS < 8, confirming the high sensitivity of the CS and its derived thresholds. Moreover, the CS threshold predicted SARS-COV-2 positivity in 100% of patients (20/170) with a first (false negative) oro-nasopharyngeal swab RT-PCR result. 

### 3.6. Correlation with Disease Severity

A post-hoc analysis was conducted, to correlate CS with COVID-19 disease severity, on all 255 COVID-19 positive patients enrolled in the study (85 patients from case-control study place 170 patients of the prospective cohort), the mean of age of the entire cohort was 68 years (±14.6), and 70.6% were male and 29.4% female. All COVID-19 patients were staged in mild, moderate, and severe according to COVID-19 Management Guidelines [19]. We considered “mild” those patients who have any of the various signs and symptoms of COVID-19 (e.g., fever, cough, sore throat, malaise, headache, muscle pain, nausea, vomiting, diarrhea, loss of taste and smell) but who do not have shortness of breath, dyspnea, or abnormal chest imaging; we considered “moderate” individuals who show evidence of lower respiratory disease during clinical assessment or imaging and who have an oxygen saturation (SpO_2_) ≥ 94% on room air at sea level; we considered “severe” individuals who have SpO_2_ < 94% on room air at sea level, a ratio of arterial partial pressure of oxygen to fraction of inspired oxygen (PaO_2_/FiO_2_) <300 mmHg, respiratory frequency >30 breaths/min, or lung infiltrates >50%. On 255 positive patients, 26.3% (67) have a mild disease, 18% (46) have a moderate disease, 49.4% (126) have a severe disease, and 6.3% (16) remain unstaged because of missing parameters. Linear non-parametric bivariate analysis using Kendall’s rank correlation (*p* < 0.01) was performed for (i) all altered blood tests, used as continuous variables, and (ii) CS versus disease severity to establish the strength of potential correlations. The results of Kendall’s correlation analysis between CS and blood tests results versus disease severity are reported in Table 5.

Cumulative score resulted to be independent from the severity of COVID-19 disease, and, in a separate analysis, it resulted to be not correlated to the number of days from symptoms onset as well (*n* = 164, correlation coefficient 0.04, two-sided *p* value 0.948), whereas a statistically significant positive correlation (*p* < 0.01) was found between severity of COVID-19 disease and values of LDH, CRP, d-dimer, PCR/Albumin, and DELR, and an inverse significant correlation was found between disease severity and albumin. 

## 4. Discussion

In this retrospective study of 85 symptomatic COVID-19 patients admitted to the ED and 85 controls with the same symptoms but confirmed as COVID-19 negative, we established a set of routine blood test parameters significantly associated with COVID-19 positivity. These parameters and their combined cumulative score might help clinicians identify potentially infected patients before formal diagnostic test results become available and to exclude infection in patients presenting with symptoms mimicking COVID-19.

Hypoalbuminemia, reduced prothrombin time percentage, and absolute and percentage lymphocytopenia and increased CRP, LDH, D-dimers—as reported in previous studies [20,21]—BUN, blood glucose levels, fibrinogen, and percentage neutrophils were, according to standard laboratory cut-offs, significantly associated with SARS-COV-2 infection on admission to the ED. Even in the absence of standardized cut-off values, there were significant differences in absolute and percentage eosinophils, WLR, NLR, and DELR between COVID-19 patients and controls using cut-offs established by ROC analysis. Eosinopenia was significantly correlated with SARS-CoV-2 infection, as reported previously [22]: an eosinophil percentage less than 0.75% and an absolute count less than 0.055 × 10^3^/μL (IS: 0.055 × 10^9^/L) was suggestive of SARS-CoV-2 infection. Similarly, a WLR > 5, an NLR > 3.76, and a DELR > 14 (ng/mL/%) were associated with ongoing infection.

A cumulative score (CS), defined as the sum of the number of all altered blood examinations in each patient, greater than 7 in patients with COVID-19 like symptoms, or in pauci-symptomatic patients with chest CT findings of bilateral interstitial pneumonia, suggested with 100% specificity SARS-CoV-2 ongoing infection, even beyond initial swab’s result. A CS less than 7 (<6.5) was highly suggestive of SARS-CoV-2 mimic conditions in symptomatic patients, with 6% of false negative results. 

In 14% of patients with a first false negative oro-nasopharyngeal swab RT-PCR result performed within 24 h of admission, 100% had a CS > 7, demonstrating how the CS might be useful in clinical practice. An independent prospective validation was performed in 170 COVID-19 patients admitted to the same ED during the “second wave” with symptoms or abnormal radiological findings suggestive of SARS-CoV-2 infection, which confirmed the high sensitivity of the CS for the rapid identification of COVID-19-positive patients. In the prospective cohort, the CS was >8 in 100% of cases, even in the 12% of patients with a first false negative RT-PCR result, confirming its high positive predictive value, even beyond RT-PCR. 

Furthermore, while, as previously reported, increasing values of PCR [23], LDH [24], d-dimer [24,25], DELR, PCR/albumin [26] and decreasing values of albumin serum level [27] at admission showed significant correlation with disease severity, CS resulted to be not correlated neither with the number of days from symptoms onset nor with disease severity itself, assessed on the basis of specific integrated criteria based on respiratory function, symptoms and radiological findings, thus reinforcing its screening power and its independent high sensibility.

This study was limited by the relatively small sample size and the variability in the number of valid results for each parameter; further studies with more data and using other predictive modeling techniques, such as binary logistic regression, might be desirable.

Moreover, the current study is limited by its exclusive applicability to symptomatic patients or to pauci-symptomatic patients who already showed chest CT scan findings suggestive of SARS-CoV-2 infection, with positive asymptomatic patients not included in the series.

## 5. Conclusions

In conclusion, a simple analysis of routine admission laboratory blood tests in the ED, together with clinical symptoms of SARS-CoV-2 infection or specific radiological findings, can help clinicians establish a pre-test (swab) risk probability of SARS-CoV-2 infection to quickly identify patients most likely to need rapid isolation, thus helping rapid triage, assessment, and management. This aspect might be considered crucial in settings of overcrowding, when the swab’s availability could be limited due to processing time and resources. Based on our results, a high pre-test diagnostic probability (CS > 7): (i) associated with an initial negative RT-PCR result suggests further testing though serial oro-nasopharyngeal swabs, broncho-alveolar lavage fluid testing, or serum immunodiagnostics in order to avoid missed diagnoses; and (ii) can be used in developing countries, where laboratory testing may be limited, for rapid triage of patients with a reasonably high sensitivity and specificity.

## Figures and Tables

**Figure 1 diagnostics-11-01566-f001:**
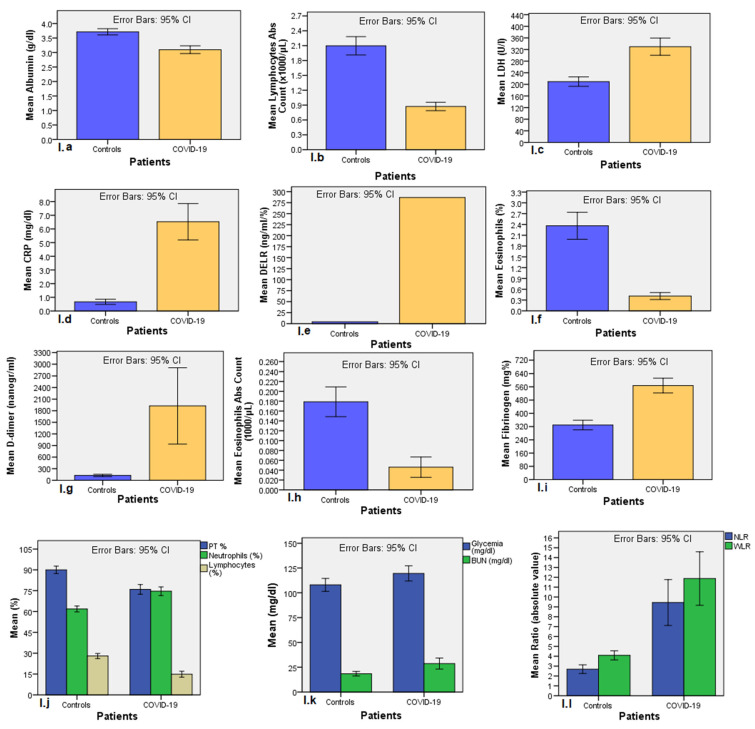
Bar charts comparing hematological parameters’ mean results between controls and COVID-19 patients. Bar errors are 95%CI (blood exams with the same unit of measurement and order of magnitude are shown in a single graph). (**a**) Albumin (n.v. 3.4–5 g/dL) (conversion factor (CF) for International System: ×10 (mg/L)); (**b**) Lymphocyte count (n.v. 0.9–5 × 10^3^/μL) (CF ×1 (10^9^/L)); (**c**) LDH (n.v. 100–240 U/L) (CF: ×0.111 (mmol/L)); (**d**) CRP (n.v. 0–1 mg/dL) (CF: ×10 (mg/L); (**e**) DELR, d-dimer to the sum of percentage lymphocytes and eosinophils; (**f**) eosinophils percentage (n.v. 0–6%) %) )CF ×0.01 (fraction of 1)); (**g**) D-dimer (n.v. 0–280 ng/mL) (CF ×1 (ng/mL)); (**h**) eosinophil count. (n.v. 0–0.6 × 10^3^/μL) (CF ×1 (10^9^/L)); (**i**) Fibrinogen (n.v. 150–400 mg/dL) (CF ×0.01 (g/dL)); (**j**) PT% (n.v. 80–120%) (CF: ×0.01 (fraction of 1)), percentage neutrophils (n.v. 40–75%) (CF ×0.01 (fraction of 1)) and percentage lymphocytes (n.v. 20–45%) (CF ×0.01 (fraction of 1)); (**k**) glucose blood levels (n.v. 74–106 mg/dL) (CF: ×0.0555 (mmol/L)) and BUN (v.n. 7–18 mg/dL) (CF: ×0.357 (mmol/L)); (**l**) NLR, neutrophils-to-lymphocytes ratio; WLR, white blood cells to lymphocytes count ratio.

**Figure 2 diagnostics-11-01566-f002:**
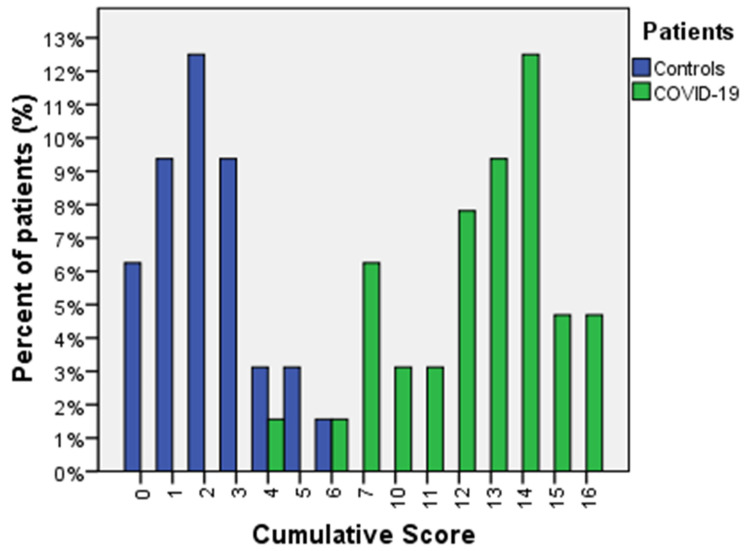
Cumulative score analysis: case-control distribution of values. Distribution of CS values between controls (green) and the COVID-19 population (blue): among controls, 13.8% had a CS of 0; 20.7% had a CS of 1; 27.6% had a CS of 2; 20.7% had a CS of 3; 6.9% had a CS of 4; 6.9% had a CS of 5; 3.4% had a CS of 6; no patient in the control group had values >6.5. Among the COVID-19 population: 2.9% had a CS of 4; 2.9% had a CS of 6; 11.4% had a CS of 7; 5.7% had a CS of 10; 5.7% had a CS of 11; 14.3% had a CS of 12; 17.1% had a CS of 13; 22.9% had a CS of 14; 8.6% had a CS of 15; 8.6% had a CS of 16. Less than 5% of COVID-19 patients had a CS < 6.5.

**Figure 3 diagnostics-11-01566-f003:**
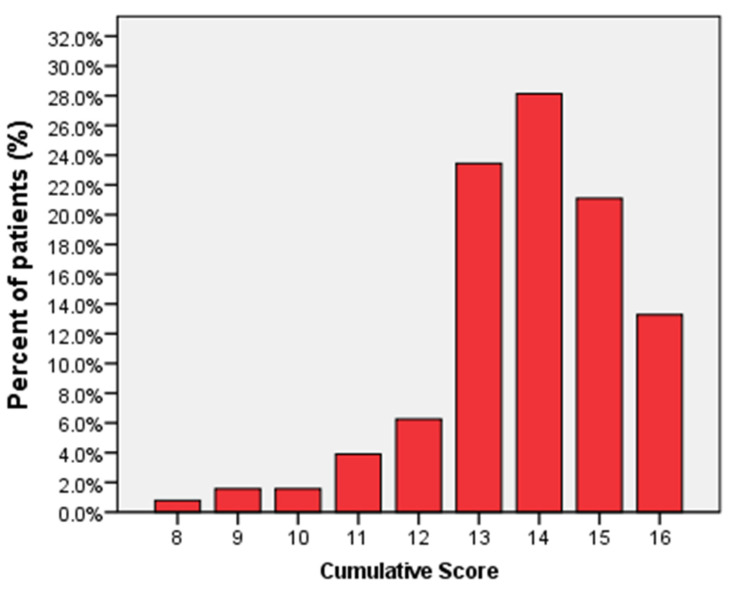
Distribution of CS values in the COVID-19 prospective cohort.

**Table 1 diagnostics-11-01566-t001:** Matched analysis between COVID-19 patients and controls and chi-squared test results for the distribution of presenting symptoms. SBP, systolic blood pressure; DBP, diastolic blood pressure; HR, heart rate (beats per minute); SpO_2_/FiO_2_, oxygen saturation level/fraction of inspired oxygen; RR, respiratory rate (breaths per minute); BT, body temperature. All radiological examinations were reported by a radiologist and carried out 24 h from admission.

Parameter	COVID-19 Group		Control Group		*p*-Value	
Age (years)	(85) 65 ± 15.3		(85) 60 ± 21		0.055	
SBP (mmHg)	(84) 132 ± 21		(85) 132 ± 27		0.387	
DBP (mmHg)	(84) 77 ± 15		(85) 76 ± 13		0.981	
HR (bpm)	(83) 89 ± 15		(85) 92 ± 20		0.125	
SpO_2_/FiO_2_	(79) 365 ± 118		(84) 395 ± 103		<0.001	
RR (bpm)	(73) 20 ± 5		(73) 18 ± 4		0.017	
BT (°C) Days from onset (*n*)	(79) 37.1 ± 0.9 (85) 6.35 ± 3.6		(82) 37.2 ± 0.9		0.359	
**Symptoms and Signs on Admission**	**COVID-19 Group** **(Frequency %)**	**Control Groups** **(Frequency %)**	**Pearson Χ2**	**df**	**Asymp. Sig (Two Sided)**	**Phi**
Dyspnea	36.5	40.0	0.224	1	0.636	−0.036
Gastrointestinal disorders	12.9	21.2	2.037	1	0.153	−0.109
Fever	25.9	34.1	1.373	1	0.241	−0.090
Non-specific symptoms with interstitial bilateral pneumonia at chest CT scan	24.7	4.7	13.553a	1	<0.001	0.282

**Table 2 diagnostics-11-01566-t002:** Physiological variables and symptoms and signs of admission of COVID-19 prospective cohort.

Parameter	COVID-19 Prospective Cohort (mean ± s.d.)
Age (years)	(170) 69 ± 14
SBP (mmHg)	(139) 127.8 ± 16.7
DBP (mmHg)	(139) 76.8 ± 11.2
HR (bpm)	(130) 88.5 ± 17.2
SpO_2_/FiO_2_	(149) 393 ± 93
RR (bpm)	(62) 20.3 ± 5.2
BT (°C)	(142) 37 ± 0.9
Days from onset	(170) 6.21 ± 2.24
**Symptoms and Signs at Admission in ED**	**COVID-19 Prospective Cohort (Frequency %)**
Dyspnoea	45.9
Gastrointestinal disorders	11.8
Fever	30.0
Non-specific symptoms with interstitial bilateral pneumonia at chest CT scan	12.4

**Table 3 diagnostics-11-01566-t003:** Frequency analysis through chi-square test of altered routine blood tests according to laboratory and ROC-established cut-points between groups. All variables were dichotomized according to standard laboratory ranges apart from NLR and WLR, which were considered abnormally increased according to ROC-established thresholds. SP, specificity; PPV, positive predictive value; NPV, negative predictive value (sensitivity corresponds to frequency for each group).

Altered Finding	Frequency (%) in COVID-19 NEG POS		Pearson χ^2^	df	2-Sided *p*-Value	φ Coeff.	SP (%)	PPV	NPV	Odds Ratio	95% CI
Hypoalbuminemia	14.3%	65.3%	39.1	1	<0.001	0.519	85.7	0.83	0.7	11.31	4.97–25.7
Hyperglycemia	28.2%	55.7%	12.72	1	<0.001	0.279	71.8%	0.65	0.64	3.19	1.67–6.11
Increased BUN	27.1%	49.4%	8.66	1	0.003	0.230	72.9%	0.63	0.61	2.63	1.37–5.04
Increased LDH	22.8%	67.9%	33.29	1	<0.001	0.452	77.2%	0.76	0.69	7.154	3.56–14.36
Increased CRP	17.3%	83.8%	71.13	1	<0.001	0.665	82.7%	0.83	0.84	24.66	10.78–56.42
Decreased PT%	19%	52.1%	18.27	1	<0.001	0.347	81%	0.72	0.64	4.63	2.24–9.57
Hyperfibrinogenemia	28.8%	84.2%	44.68	1	<0.001	0.561	71.2%	0.77	0.8	13.19	5.84–29.80
Increased D-dimers	8.3%	69.4%	31.47	1	<0.001	0.608	91.7%	0.91	0.69	24.93	6.6–94.17
Neutrophilia (%)	8.3%	58.8%	47.16	1	<0.001	0.536	91.7%	0.87	0.7	15.67	6.42–38.25
Lymphocytopenia (%)	10.8%	73.4%	65.35	1	<0.001	0.635	89.2%	0.87	0.78	22.71	9.67–53.3
Eosinopenia (%)	9.5%	86.1%	95.87	1	<0.001	0.767	90.5%	0.89	0.87	58.72	22.31–154.6
Lymphocytopenia (Absolute Count)	2.4%	55.1%	56.1	1	<0.001	0.588	97.6%	0.95	0.70	50.37	11.56–219.5
Eosinopenia (Absolute Count)	9.5%	81.5%	86.32	1	<0.001	0.723	90.5%	0.89	0.84	41.8	16.67–104.8
Increased NLR	10.7%	74%	65.05	1	<0.001	0.644	89.3%	0.86	0.8	23.68	9.95–56.35
Increased WLR	11.9%	76.7%	67.32	1	<0.001	0.655	88.1%	0.84	0.81	24.37	10.37–57.30
Increased DELR	0%	84.8%	57.23	1	<0.001	0.841	100%	1	0.83	>190	-

**Table 4 diagnostics-11-01566-t004:** Descriptive statistics of blood tests in the COVID-19 prospective cohort. WLR, white blood cell to lymphocyte count; DELR, d-dimer to (eosinophils% + lymphocytes%).

	*n*	Mean	SD	Variance	Skewness		Kurtosis	
					Statistic	SE	Statistic	SE
Albumin (g/dL)	167	2.919	0.4838	0.234	0.056	0.188	−0.040	0.374
Glycemia (mg/dL)	169	158.01	87.784	7706.095	2.739	0.187	9.834	0.371
BUN (mg/dL)	169	29.64	17.823	317.673	1.812	0.187	3.348	0.371
LDH (U/L)	165	395.12	135.423	18,339.395	0.727	0.189	0.351	0.376
CRP (mg/dL)	167	10.331	7.2912	53.162	1.192	0.188	1.817	0.374
PT%	166	77.76	13.905	193.361	−1.544	0.188	6.451	0.375
Fibrinogen (mg%)	156	657.53	193.573	37,470.548	0.503	0.194	0.469	0.386
D-dimer (ng/mL)	151	1699.80	4982.220	24,822,520	6.016	0.197	41.908	0.392
Neutrophils (%)	169	85.180	6.5578	43.005	−0.899	0.187	0.847	0.371
Lymphocytes (%)	169	8.482	4.6368	21.500	0.860	0.187	0.035	0.371
Eosinophils (%)	169	0.299	0.3773	0.142	2.638	0.187	9.158	0.371
Lymphocytes Count (×1000/μL)	169	0.7021	0.31610	0.100	1.194	0.187	1.531	0.371
Eosinophil count (×1000/μL)	169	0.0289	0.04104	0.002	2.729	0.187	8.011	0.371
NLR	169	14.0779	9.48520	89.969	2.017	0.187	7.909	0.371
WLR	169	16.0802	9.82704	96.571	1.956	0.187	7.606	0.371
DELR	146	272.6488	900.68107	811,226.399	7.517	0.201	64.362	0.399

**Table 5 diagnostics-11-01566-t005:** Non-parametric bivariate correlation between Cumulative Score and blood tests results versus COVID-19 disease severity.

Parameter	*n*	Kendall’s Tau B Correlation Coefficient	Sig. (Two-Tailed)
COVID-19 disease severity	239	1	
ays from onset	239	0.069	0.194
CS	147	0.126	0.071
Albuminemia (g/dL)	226	−0.140	<0.01
Glucose blood levels (mg/dL)	232	0.002	0.973
BUN (mg/dL)	232	0.061	0.242
LDH (U/L)	233	0.221	<0.01
CRP (mg/dL)	231	0.165	<0.01
PT percentage (%)	223	−0.60	0.257
Fibrinogen (mg%)	216	0.077	0.147
D-dimer (nanogr/mL)	184	0.171	<0.01
Neutrophils percentage (%)	233	0.038	0.457
Lymphocyte percentage (%)	232	−0.062	0.227
Eosinophil percentage (%)	232	−0.076	0.163
Lymphocyte count (× 10^3^/μL)	231	−0.072	0.166
Eosinophil count (× 10^3^/μL)	232	−0.076	0.163
NLR	226	0.081	0.120
WLR	226	0.080	0.122
DELR	176	0.158	<0.01
CPR/Alb	222	0.167	<0.01

## Data Availability

All data are presented within this manuscript.

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
