# Peer review of "A Diagnostic Model to Predict SARS-CoV-2 Positivity in Emergency Department Using Routine Admission Hematological Parameters"

_diagnostics, 2021, doi:10.3390/diagnostics11091566_

Round 1

Reviewer 1 Report

Authors presented an interesting retrospective case control study named: A diagnostic model to predict SARS-CoV-2 positivity in Emergency Department using routine admission hematological parameters.

Few remarks: 

  1. Supplementary- tables (S1, S2, S3, S4) are too extensively and too unnecessary explained in the title part. Repetitions from tables, tables itself is enough, no need to repeat all lab results in title.
  2. Materials – chest CT scan was performed to diagnose all pneumonia, and not standard chest X-rays? Can you explain?

Furthermore, did control patients also receive chest CT scan, and what are their results (inclusion criteria part).

  1. Can you list the possible bias and confounding factors? Especially in inclusion/exclusion part? I feel that you have been very “picky” about choosing the patients.

Author Response

Dear Reviewer,
first of all, we sincerely thank you for taking the time to assess our manuscript.
We have carefully read your revision and addressed all the concerns you raised. We report our reply to your revisions point by point (please see in black your comment and in red ours), in order.
1. Supplementary- tables (S1, S2, S3, S4) are too extensively and too unnecessary explained in the title part. Repetitions from tables, tables itself is enough, no need to repeat all lab results in title.7
We thank you for pointing this out and apologize for our error. We revised the title parts you cited, deleting what was unnecessary. To better present our results and divide case-control study from prospective study, we made same changes in figure presentation, please find:
• Table S1 as Table 3 (please see page 6 in the revised manuscript), which has been expanded with vital parameters of the prospective cohort for better description of the population in exam. For this table we eliminated in the title part all the superfluous.
• Table S2 and S3 as Table S1: we appreciate your comment and agree that it would be better fix the error, however, for this table unfortunately we have not had a huge possibility to synthesize because it’s the first part in which all blood tests are presented, we have taken steps to eliminate the superfluous, but we had to keep just the specification of the acronyms right.
• Table S4 as Panel 2 (I) (please see page 11 in the manuscript): we eliminated all the repetitions and what was unnecessary.
2. Materials – chest CT scan was performed to diagnose all pneumonia, and not standard chest X-rays? Can you explain?
We appreciate your insightful suggestion and agree, we integrated our manuscript with further details to better describe our clinical choice, please see page 4 of the revised manuscript, lines 181-187. We have revised the text to address your concern and hope that is now clearer. We also added in the manuscript (please see page 4 lines 190-194) the peculiar radiological patterns of Covid-19 disease and delete this description from title part of Table 1.
We used chest X-rays and/or lung ultrasound for all patients admitted in ED for suspicious symptoms. A second level imaging with chest-CT scan was then systematically performed in all COVID-19 positive patients before department admission in order to “stage” them on the bases of radiological patterns, that means: (i) to individuate patients who did not present any chest radiological involvement and address them vs low intensity care units, and, (ii) among patients with positive chest CT scan, to differentiate between “L” and “H” phenotype, as described by Prof. Gattinoni (reference n.18 in the article), and consequently use proper non-invasive/ invasive ventilation.
Furthermore, did control patients also receive chest CT scan, and what are their results (inclusion criteria part).
Among control patients, only those patients for which chest x-rays and lung ultrasounds were not sufficient for definitive diagnosis underwent to chest-CT scan, there were no overlaps with COVID-19 radiological patterns apart for those control patients having an interstitial atypical pneumonia or alveolitis, in which we found similarities; we update our manuscript with this information (please see page 4 of the revised manuscript, lines 187-190). We also added information about the aetiology of admission in ED of control group (please see page 5, Table 2).
3. Can you list the possible bias and confounding factors? Especially in inclusion/exclusion part? I feel that you have been very “picky” about choosing the patients.
Thank you for your comment, actually inclusion and exclusion criteria in the original manuscript were not adequately explained, we did our best to integrate this part and hope now is clearer: please see page 2 of the revised manuscript, lines 72-77 and 83-90 for the integrated inclusion/exclusion criteria. We have revised the text to address your concern and integrating in the manuscript the possible bias and confounding errors not mentioned before (please see in the revised manuscript
page 2, lines 90-95); moreover, we added information about excluded patients in lines 170-181 (please see page 4 of the revised manuscript), and we added a patient flowchart (Figure S1).
The process of eligibility of positive patients, in case-control study, started by collecting patients discharged from ED with a diagnosis of “COVID-19 disease” (N=138). Unfortunately, when we retrospectively analysed every single patient, we found that for the 25,36% of them (35 patients on 138) the diagnosis of COVID-19 was actually “suspected” rather than “proved” (because of doubtful results of swabs performed in ED or, mostly, because of the impossibility to obtain a certain diagnosis in patients died in ED at admission, before performing swab). Even if they were very likely to be COVID-19 patients, according to symptoms, arterial blood gas and radiological imaging, we had to exclude them from our database because missing certain diagnosis. While, 18/138 (13,04%) patients were excluded for severe pre-existing conditions highly impacting on haematological blood exams, mostly terminal lymphoma or leukaemia, not for a matter of principle but after data analysis, in which we found their blood tests presented extreme abnormal values which would have created outliers and confounding errors. The same selection process was applied for the prospective cohort.
For controls patients, the enrolment process started from patients officially discharged from ED with “other diagnosis Covid free”, presented at admission with suspicious symptoms (N=102). We had to exclude all patients presented with negative swab and positive serum IgM/IgG (swab false negative result? Second phase of COVID-19 disease?) even if clinically diagnosed as COVID-19 negative, because we could not consider those patients totally COVID-free for sure (8/102), and patients presented with very abnormal blood tests due to other conditions for the reason explained above (9/102).
We would like to thank you again to taking the time to review our manuscript and thank you un advance for your attention.
Best regards,
Giovanna Guiotto

Reviewer 2 Report

This is an interesting manuscript reporting on a predictive model based on blood parameters to screen for SARS-CoV-2.

Major remarks

  • For the retrospective case-control study, 85 symptomatic SARS-CoV-2-negative controls were eligible. Have the authors some information about the etiology of their symptoms? The specificity of the diagnostic model for SARS-CoV-2 should be tested versus other infectious, metabolic or inflammatory diseases that can modify hematological and biochemical parameters.

  • SARS-CoV-2-positive patients should be classified according to disease severity (mild, moderate or severe disease). Does the sensitivity of the current diagnostic model change according to disease severity? Information about time after symptoms’ onset should be also indicated as blood parameters may fluctuate over time in an infected individual.

  • The results should be better organized in order to clearly distinguish the retrospective case-control study and the validation study. I suggest:
  • For the retrospective case-control study: to pool tables 1 (physiological variables) and 2 (symptoms) in one table. To move figure S1 (the sixteen parameters significantly different between the two groups) in the manuscript and associate it with table 3 and figure 2a (the cumulative score). To move table 4 in supplemental results. Of note, table S2 (biochemical and not “hematological” parameters) and table S3 (hematological parameters) show more than sixteen significantly altered parameters in the COVID-19 group. The authors should justify why they decided to focus on the sixteen parameters described in the manuscript.
  • For the validation study: to move table S1 in the manuscript and to complete it with physiological variables. To move table S4 in the manuscript and to show, in the same panel, figure 2c (the cumulative score for the prospective COVID-19 cohort).
  • ROC curve analysis used to determine cut-off for non-standard parameters should be moved in supplemental data.

Minor remarks

  • Abstract, lines 30-33: the authors describe the cumulative score as a way “to provide an early and accurate diagnosis of SARS-CoV-2 infection in symptomatic patients (…)” In my opinion, the term “to provide” is not appropriate and should be substituted by another one such as “to predict”.

  • A patient flow-chart could be inserted in the manuscript.

Author Response

Dear Reviewer,
first of all, we sincerely thank you for taking the time to assess our manuscript.
We have carefully read your revision and addressed all the concerns you raised. We report our reply to your revisions point by point (please see in black your comment and in red ours), in order.
Major remarks
• For the retrospective case-control study, 85 symptomatic SARS-CoV-2-negative controls were eligible. Have the authors some information about the etiology of their symptoms? The specificity of the diagnostic model for SARS-CoV-2 should be tested versus other infectious, metabolic or inflammatory diseases that can modify hematological and biochemical parameters.
• We really thank you for pointing this out, we didn’t realize this information was important. We improved our manuscript adding the aetiology of admission of controls - retrospectively found in our online patients’ platform - in a new specific table (please see page 5 of the revised manuscript, Table 2, line 216) with all the information required.
• SARS-CoV-2-positive patients should be classified according to disease severity (mild, moderate or severe disease). Does the sensitivity of the current diagnostic model change according to disease severity? Information about time after symptoms’ onset should be also indicated as blood parameters may fluctuate over time in an infected individual.
• We appreciate your insightful and agree it’s useful to elaborate this point introducing new data. In order to establish if a correlation with disease severity was present, we put together all positive patients enrollable for the analysis (85 COVID-19 positive patients from case-control study place 170 patients of the prospective cohort) and staged them, according to Covid-19 Management Guidelines (reference n.19 in the manuscript), by obtaining three classes of patients: mild, moderate and severe (where not present in our analysis pre-clinical disease because no asymptomatic patients was enrolled in our study). We then performed a linear non-parametric correlation between severity of disease and CS: the correlation resulted to be not statistically significant (Kendall’s tau B correlation coefficient 0.126, two-tiled P value 0,071).
Once we staged patients for disease severity, we also correlated their blood tests results for severity, founding those haematological parameters (considered as continuous variables) correlated with the stage of the disease, as it was reported in previous (prognostic) studies. You’ll find the paragraph regarding disease severity assessment, with all the criteria followed, and the correlation analysis in page 12 of the revised manuscript, lines 535-556 and 607-613, moreover, we summarize the results of non-parametric correlation in Table 4 (page 12) in the manuscript. We consequently improved the bibliography of the study adding the corresponding references (references 19, 23-27). Please see pag.3 lines 148-155 in which we added this post-hoc analysis in the paragraph Statistical Analysis and line 30 in which we added this results in the Abstract.
As you suggested, we also analysed the correlation between the number of days from symptoms onset with the severity of disease (showed in Table 4), and, in a separate correlation, not showed in the table but reported in the manuscript (page 13, lines 608-609), we performed the same analysis considering CS vs days from symptoms onset. Both these correlations resulted to be not statistically significant.
We concluded that CS is not correlated, according to our data, neither with disease severity nor with the number of days from symptoms onset. Moreover, we concluded that values of LDH, CRP, d-dimer, PCR/Albumin ratio, and DELR significantly increase with the severity of disease, while values of serum albumin tend to decrease with the severity of COVID-19 disease.
Please see page 16 lines 649-655 when we reported these results in “Discussion”.
• The results should be better organized in order to clearly distinguish the retrospective case-control study and the validation study. I suggest:
For the retrospective case-control study: to pool tables 1 (physiological variables) and 2 (symptoms) in one table. To move figure S1 (the sixteen parameters significantly different between the two groups) in the manuscript and associate it with table 3 and figure 2a (the cumulative score). To move table 4 in supplemental results.
We thank you very much for this improvement. We agree with you that this presentation of results better divide the two parts of the manuscript, and we apologize if before the distinction was not so clear.
For the retrospective case-control study, as you suggested, we pooled table 1 and table 2 in one table (Table 1, please see page 4 of the revised manuscript); we moved figure S1 in the manuscript and associated it with table 3 and figure 2a (please see pages 8-10, PANEL1: (I), (II), (III), respectively), that we put at the end of case-control study to give and immediate summarize of results; we moved table 4 in supplementary material (please see Table S2).
Of note, table S2 (biochemical and not “hematological” parameters) and table S3 (hematological parameters) show more than sixteen significantly altered parameters in the COVID-19 group. The authors should justify why they decided to focus on the sixteen parameters described in the manuscript.
We apologies if this information was not adequately stressed before, we corrected. We specified into “Materials and Methods” section (please see page 3, lines 135-139) the reasons for which not all the haematological parameters which resulted significantly different between COVID-19 group and controls were considered for further analysis. We reproposed the concept at paragraph 3.2 (“Univariate differences in haematological parameters between SARS-CoV-2-positive and negative patients (retrospective case-control study”) at lines 237-242 (page 7), immediately after the presentation of data of the complete univariate analysis (now Supplementary Table 1), in order to recall this concept during manuscript’s reading. We also pooled Table S1 and S2 in one table (now Table S1) because they were linked.
To better explain our decision, i.e., Calcium (mg/dl) mean for control patients was 8.89 ± 0.45, while calcium mean for COVID-19 patients was 8.49 ± 0.62, the difference was statistically significant (P < 0.001) but the distribution analysis of calcium values showed that, both in COVID-19 and control patients, calcium values felt into normality range (8.5-10.1 mg/dl). In these cases, the variable was excluded from further analysis because useless in clinical practice. The only exception was “eosinophils”, for which the difference between the two groups both in terms of mean and distribution of values was huge and net, that’s why we proceed in the analysis and decide to create a non-standard cut-off through ROC curve.
Were also excluded those values resulted significantly different but whose alteration (according to laboratory cut-off) was found in less than 50% of COVID-19 patients, to establish a cut off of sensibility, retaining unluckily to be useful, in a screening test, parameters with low sensibility.
For the validation study: to move table S1 in the manuscript and to complete it with physiological variables. To move table S4 in the manuscript and to show, in the same panel, figure 2c (the cumulative score for the prospective COVID-19 cohort).
For the validation study, we moved table S1 in the manuscript, as you suggested (please see page 6 of the revised manuscript, Table 3) and completed it with prospective cohort’s physiological variables, as requested (please see page 6 lines 227).
We then moved table S4 in the manuscript and associated it with figure 2c (please see page 11 of the revised manuscript, PANEL 2: (I), (II), respectively).
• ROC curve analysis used to determine cut-off for non-standard parameters should be moved in supplemental data.
• We have made the change and moved all ROC curve analysis in the supplementary material.
.
Minor remarks
• Abstract, lines 30-33: the authors describe the cumulative score as a way “to provide an early and accurate diagnosis of SARS-CoV-2 infection in symptomatic patients (…)” In my opinion, the term “to provide” is not appropriate and should be substituted by another one such as “to predict”.
• We thank you for your suggestion and apologize for our error, we deleted “provide” and replaced it with “predict”, more appropriate (please see page 1 line 31).
• A patient flow-chart could be inserted in the manuscript.
• We agree with this observation. We added a patient flow-chart in Figure 1 in Supplementary material. We think this visual representation helps to focalize the study design and thank you for improvement.
We would like to thank you again to taking the time to review our manuscript and thank you un advance for your attention.
Best regards,
Giovanna Guiotto

Round 2

Reviewer 2 Report

The authors have satisfactorily addressed all my comments and I would like to thank them for this revision.

In my opinion, the manuscript could be accepted after these last minor corrections:

  • Table 2 (etiology of admission in ED of control group) should be moved in in supplemental data.
  • Please correct “is summarize” by “is summarized” at line 199.

Author Response

Dear Reviewer,
we again thank you for taking the time to assess our manuscript.
We addressed all the concerns you raised. Here in order our reply to your comments: 

  • Table 2 (etiology of admission in ED of control group) should be moved in in supplemental data
  • As you suggested, we moved Table 2 in Supplementary Material, and consequently updated the numbering of other images, without any other change. Please see (old version vs revised):

         Table 2 --> Table S1

Table 3 --> Table 2 

Table S1 --> Table S2 

Table S2 --> Table S3

Table 4 --> Table 3

  • Please correct “is summarize” by “is summarized” at line 199.
  • We thank you for the correction, we fixed it by replacing "summerize" with "summerized" (please see page 4 of Manuscript, line 183).

We sinceraly thank you for all your precious comments, which significantly improved our work.

Best regards, 

Giovanna Guiotto